# Genome-wide analysis of gibberellin-dioxygenases gene family and their responses to GA applications in maize

**Jiabin Ci[1], Xingyang Wang[1], Qi Wang[1], Fuxing Zhao[1], Wei Yang[1], Xueyu Cui[2], Liangyu Jiang[1], Xuejiao Ren[1], Weiguang Yang[1]***

**1** College of Agronomy, Jilin Agricultural University, Changchun, China, **2** Key Laboratory of Beibu Gulf Environment Change and Resources Utilization of Ministry of Education, Nanning Normal University, Nanning, China

\* ywg798@126.com

**Data Availability Statement:** All relevant data are within the manuscript and its Supporting Information files.

## Abstract

Gibberellin-dioxygenases genes plays important roles in the regulating plant development. However, Gibberellin-dioxygenases genes are rarely reported in maize, especially response to gibberellin (GA). In present study, 27 Gibberellin-dioxygenases genes were identified in the maize and they were classified into seven subfamilies (I-VII) based on phylogenetic analysis. This result was also further confirmed by their gene structure and conserved motif characteristics. And gibberellin-dioxygenases genes only occurred segmental duplication that occurs most frequently in plants. Furthermore, the gibberellin-dioxygenases genes showed different tissue expression pattern in different tissues and most of the gibberellin-dioxygenases genes showed tissue specific expression. Moreover, almost all the gibberellin-dioxygenases genes were significantly elevated in response to GA except for *ZmGA2ox2* and *ZmGA20ox10* of 15 gibberellin-dioxygenases genes normally expressed in leaves while 10 and 11 gibberellin-dioxygenases genes showed up and down regulated under GA treatment than that under normal condition in leaf sheath. In addition, we found that *ZmGA2ox1*, *ZmGA2ox4*, *ZmGA20ox7*, *ZmGA3ox1* and *ZmGA3ox3* might be potential genes for regulating balance of GAs which play essential roles in plant development. These findings will increase our understanding of Gibberellin-dioxygenases gene family in response to GA and will provide a solid base for further functional characterization of Gibberellin-dioxygenases genes in maize.

## Introduction

The development of plant organs is directly dependent on the frequency of cell division, the parameters of the cell cycle, and the number and size of the cells [1]. Plants are continuously exposed to a variety of stress factors in their natural environment. Of them, gibberellins (GAs) play multiple roles in plant development and stress responses which will significantly affect the production and quality of the plants [2, 3]. To adapt natural environment, plants have to

**Funding:** The work was carried out with the financial support of Science and Technology Development Plan of Jilin Province (20190301013NY); The National Key Research and Development Program of China (2016YFD0101202).

**Competing interests:** The authors have declared that no competing interests exist.

acclimate to GA by triggering a cascade of events leading to changes in gene expression and subsequently to biochemical and physiological modifications. GA synthesis, metabolism and GA signaling transduction play core roles to cope with various natural environment. However, although much effort, the key genes and signaling pathways involved in GA remains need for further study.

With development of advanced technologies, numbers of genes which contribute to GA signaling were discovered. Genetic analyses of GA-deficient and GA-response mutants have revealed that the central step in GA action is to turn off the repressive effects of DELLAs in plants. In the presence of GA, the GA-GID1-DELLA complex stimulates the interaction of DELLAs with an F-box protein, resulting in the degradation of DELLAs and consequently the activation of downstream-responsive processes [4]. In higher plants, the flux of active GAs is regulated by the balance between their rates of biosynthesis and deactivation. The GA 20-oxidase (GA20ox) and GA 3-oxidase (GA3ox) genes encode key enzymes of bioactive GAs synthesis, whereas GA 2-oxidase (GA2ox) is the major GA inactivation enzyme [5]. In fact, increasing numbers of studies have investigated the gibberellin oxidase gene family in various kinds of plants, such as rice, Arabidopsis, soybean, Grape and Phyllostachys edulis [6–8]. In addition, the function of several gibberellin-dioxygenases genes has been clarified. For example, Shan et al., demonstrated that OsGA2ox5 was involved in plant growth, the root gravity response and salt stress [9]. Gibberellin 20-oxidase promoted initiation and elongation of cotton fibers by regulating gibberellin synthesis [10] while Gibberellin 20-Oxidase dictated the flowering-runnering decision in Diploid Strawberry [11]. And overexpression of jatropha gibberellin 2-oxidase 6 (jcga2ox6) induced dwarfism and smaller leaves, flowers and fruits in Arabidopsis and Jatropha [12].

Maize is one of the most important cereal crops worldwide. GA has been showed play essential roles in response to environment stress during the development of maize. Yang et al., demonstrated that GA could improve the resistance of tebuconazole-coated maize seeds to chilling stress by microencapsulation [13]. Hu et al., found that GA promote brassinosteroids action and both increase heterosis for plant height [14] and Chen et al., considered that dwarfish and yield-effective GM maize could be developed through passivation of bioactive gibberellin [15]. Recently, Zhang and Wang demonstrated that GA signaling play important roles in response to phosphate deficiency and nitrogen uptake, respectively [16, 17]. In addition, increasing numbers of studies have demonstrated that numbers of genes involved in GA signaling which contribute to the development and the production of maize. For example. Wang et al., (2013) provided physiological and transcriptomic evidence that gibberellin biosynthetic deficiency was responsible for maize dominant dwarf11 (d11) mutant phenotype and they found that the expression of ent-kaurenoic acid oxidase (KAO), GA20ox and GA2ox are up-regulated in D11 [18]. Recently, some GA-responsive transcripts which encoded the components of GA pathway were showed differential expressed in wild type and D11 in response to gibberellin stimulation, including CPS, KS, and KO enzymes for GA biosynthesis, GA2ox enzymes for GA degradation, DELLA repressors and GID1 receptor for GA signaling [19]. Muylle et al., demonstrated that overexpression of GA20-OXIDASE1 impacts plant height, biomass allocation and saccharification efficiency in maize [20].

Taken together, these results demonstrated that the biosynthesis and deactivation of gibberellin-dioxygenases genes played essential roles in maize involved in GA induced growth and development. However, there is few systematic and complete investigation on gibberellin-dioxygenases genes family in maize. Therefore, in present study, we aimed to investigate the characteristics of the biosynthesis and deactivation of gibberellin-dioxygenases gene family and identify the key genes in response to GA in maize.

## Materials and methods

### Plant materials and GA treatments

Seeds of the maize (Zea mays L.) are disinfected with 2% sodium hypochlorite (NaClO) or 70% ethanol and then rinsed with distilled water three times. And the seeds then were grown in a greenhouse at 28˚C/23˚C(day/night) with a 16-h light/8-h dark photoperiod. For gibberellin (GA) treatment, seedlings were treated with 150mg/L GA with spraying to the leaves at two leaves and one heart period. During the period of GA treatment, the seedlings were watered every day, and control seedlings were maintained under non-stress conditions. After treatment for 6, 12, 24, 48, 72 h, the samples were collected and immediately frozen in liquid nitrogen and stored at -80˚C for RNA isolation. The seedlings without GA treatment at 0 h act as control. There were three biological replicates for each experiment.

### Identification of gibberellin-dioxygenases genes in maize

The Hidden Markov Model (HMM) profile of gibberellin-dioxygenases gene (accession number PF03171.20) was downloaded from the Pfam database (http://pfam.xfam.org/). All gibberellin-dioxygenases genes were obtained by screening protein sequences of maize using HMMER 3.0 software (http://hmmer.janelia.org/) and blastp (National Center for Biotechnology Information (NCBI) Basic Local Alignment Search Tool, e-value < = 0.001). The putative gibberellin-dioxygenases genes were checked by the NCBI Conserved domain database (CDD) and Simple Modular Architecture Research Tool (SMART) online. The gibberellin-dioxygenases genes from Arabidopsis and rice were download from TAIR (Arabidopsis Information Resource, https://www.arabidopsis.org/) and Rice Genome Annotation Project Database (http://rice.plantbiology.msu.edu/ respectively).

### Characteristics of gibberellin-dioxygenases genes in maize

Both genome and coding sequences of gibberellin-dioxygenases genes were downloaded from the whole genome of maize (B73-REFERENCE-GRAMENE-4.0) database (https://alpha.maizegdb.org/). For gene structure analysis, genomic and CDS sequences were used for drawing gene structure schematic diagrams with the Gene Structure Display Server from the Center for Bioinformatics at Peking University (http://gsds.cbi.pku.edu.cn/index.php). Isoelectric point (PI) and Molecular weight (MW) of the gibberellin-dioxygenases proteins were analyzed by EXPASY website tool (https://web.expasy.org/compute_pi/). The map of the chromosome location with genes was constructed through the online software MapGene2Chrom web v2. Species-wide gene replication events was performed by using MCScanX.

### Conserved motif distributions and phylogenetic analysis

Conserved motifs for each gibberellin-dioxygenases amino acid sequence were analyzed by Multiple Em for Motif Elicitation online software (MEME, http://meme-suite.org/tools/meme). Amino acid sequences of gibberellin-dioxygenases genes were used to build the phylogenetic tree. Prottest was firstly use to predict the best evolution model and JTT+G+I+F as the best evolution model to build the evolution tree using RAxML 1000 bootstrap and the phylogenetic tree visualization is done using Figtree software.

### Tissue specific and GA induced expression analysis in maize

RNA-Seq datasets for tissue and GA treatment were downloaded from the NCBI sequence read archive (SRA) database (PRJNA314400 and PRJNA421076, respectively) [19, 21], then used to analyze the expression profiles of the identified gibberellin-dioxygenases genes. A total

of 23 tissues spanning vegetative and reproductive stages of maize development, as well as GA treatments were used to identify tissue-specific or GA responsive ones. Trimmomatic was used to remove the sequencing adapters and low-quality reads; Clean reads were aligned to the reference genome by Hisat2 and Htseq was used to calculate the counts of the reads that aligned to the genome. And TPM was used to homogenize the gene expression data. After the expression data of tissue expression in maize is transformed by zscore, it is displayed on the iTOL online tool together with the motif information; The differential expressed gene analzed by using DESeq2.

## Quantitative reverse transcription polymerase chain reaction (qRT-PCR)

Total RNA was extracted from tissues by using RNAprep pure Plant Kit (DP432, TIANGEN). 2 ug RNA was used to synthesize cDNA using PrimeScript™ RT reagent Kit with gDNA Eraser (RR047A, Takara, Japan) according to the manufacturer instructions. qRT-PCR was performed using ABI 7500 instrument (ABI7500, ABI, Foster City, CA, USA) with Geneseed® qPCR SYBR® Green Master Mix (Geneseed) with 20 μL reaction mixture of volume. The reaction volume consists of 10 μL SYBR® Green Master Mix, 0.5 μL of each primer (10 μM), 2 μL of the cDNA template, and 8 μL of RNase free $H_2O$. Thermal cycling parameters for the amplification were as follows: 95˚C, 5 min, followed by 40 cycles at 95˚C,10 s and 60˚C, 34 s. The expression level of gibberellin-dioxygenases genes were calculated by $2^{-\triangle\triangle Ct}$ methods. Actin act as internal reference. Primers used in the present study were synthesized by BGI and the detailed information was listed in **S1 Table**.

## Statistical analysis

All the data from more than three biological repeats was analyzed using the SPSS 21.0 (SPSS, Inc., Chicago, IL, USA) software. Quantitative data was presented as mean ± SD. The significance of differences between normal group and GA treatment group were assessed by the paired t test. Significant differences were finally defined as $P < 0.05$.

## Results

### Identification of gibberellin-dioxygenases genes in maize

Based on the genome and transcriptome databases, candidate gibberellin-dioxygenases genes were explored through searching against genome of maize using HMMSearch (PF03171.20) and BLASTP (e-value $< = 0.001$) methods. Totally, 38 candidate gibberellin-dioxygenases were obtained in maize. After removing redundant sequences and confirming the presence of gibberellin-dioxygenases domains by MEME, 27 Gibberellin-dioxygenases genes were finally retained and used for further analysis, including 13 GA2ox1 genes (*ZmGA2ox1-13*), 11 GA20ox genes (*ZmGA20ox1-11*) and 3 GA3ox (*ZmGA3ox1-3*) genes, respectively. Further analysis showed that these gibberellin-dioxygenases genes varied from 903 (*ZmGA20ox4*) to 1392 (*ZmGA20ox10*) nucleic acid in length (**Table 1**) and the exon numbers were 0 or 3 (**Fig 1**). Their molecular weight ranged from 32.3 kDa (*ZmGA20ox4*) to 50.7 kDa (*ZmGA20ox10*) and the PI ranged from 5.1 (*ZmGA2ox5*) to 8.91 (*ZmGA2ox9*), suggesting that 37 Gibberellin-dioxygenases might play different roles involved in different processes in maize (**Table 1**).

### Chromosome distribution of gibberellin-dioxygenases genes in maize

Generally, genes often undergo replication events during evolution. In order to know whether gibberellin-dioxygenases genes also experienced gene replication events, the chromosome distribution of 27 gibberellin-dioxygenases genes were analyzed. The results showed that these

**Table 1. Characteristic of gibberellin-dioxygenases genes in maize.**

| geneName | Gene | Transcript | Chrom | Start | End | Strand | Length of CDS | Length of peptide | PI | MW |
|---|---|---|---|---|---|---|---|---|---|---|
| ZmGA2ox1 | Zm00001d002999 | Zm00001d002999_T001 | 2 | 29175293 | 29178113 | + | 1089 | 362 | 6.66 | 38914.2 |
| ZmGA2ox1 | Zm00001d002999 | Zm00001d002999_T002 | 2 | 29175294 | 29176187 | + | 420 | 139 | 8.77 | 14767.8 |
| ZmGA2ox2 | Zm00001d039394 | Zm00001d039394_T001 | 3 | 3274307 | 3276532 | + | 996 | 331 | 5.83 | 35103.8 |
| ZmGA2ox3 | Zm00001d040737 | Zm00001d040737_T001 | 3 | 60952847 | 60960982 | - | 1050 | 349 | 8.54 | 36939 |
| ZmGA2ox4 | Zm00001d043411 | Zm00001d043411_T001 | 3 | 199019233 | 199021284 | - | 996 | 331 | 8.3 | 35497.3 |
| ZmGA2ox5 | Zm00001d017294 | Zm00001d017294_T001 | 5 | 192007580 | 192011428 | - | 1086 | 361 | 5.1 | 39407.4 |
| ZmGA2ox5 | Zm00001d017294 | Zm00001d017294_T002 | 5 | 192009900 | 192011331 | - | 579 | 192 | 4.43 | 20599.7 |
| ZmGA2ox6 | Zm00001d035994 | Zm00001d035994_T001 | 6 | 65164556 | 65165566 | - | 1011 | 336 | 6.13 | 35937.5 |
| ZmGA2ox7 | Zm00001d037565 | Zm00001d037565_T001 | 6 | 129802297 | 129807428 | - | 618 | 205 | 6.23 | 22653.8 |
| ZmGA2ox7 | Zm00001d037565 | Zm00001d037565_T002 | 6 | 129802305 | 129807398 | - | 1101 | 366 | 8.83 | 39675.8 |
| ZmGA2ox7 | Zm00001d037565 | Zm00001d037565_T003 | 6 | 129802324 | 129807396 | - | 618 | 205 | 6.23 | 22653.8 |
| ZmGA2ox8 | Zm00001d037724 | Zm00001d037724_T001 | 6 | 135240118 | 135243717 | + | 453 | 150 | 11.5 | 15765.7 |
| ZmGA2ox8 | Zm00001d037724 | Zm00001d037724_T002 | 6 | 135240119 | 135243717 | + | 1002 | 333 | 8.22 | 35995 |
| ZmGA2ox8 | Zm00001d037724 | Zm00001d037724_T003 | 6 | 135242721 | 135243667 | + | 213 | 70 | 10.8 | 8108.28 |
| ZmGA2ox9 | Zm00001d038695 | Zm00001d038695_T001 | 6 | 162656616 | 162658442 | + | 1023 | 340 | 8.91 | 36029.1 |
| ZmGA2ox10 | Zm00001d038996 | Zm00001d038996_T001 | 6 | 168446326 | 168448293 | - | 1122 | 373 | 8.8 | 39549.3 |
| ZmGA2ox11 | Zm00001d008909 | Zm00001d008909_T001 | 8 | 24852743 | 24853978 | - | 210 | 69 | 5.39 | 7392.33 |
| ZmGA2ox11 | Zm00001d008909 | Zm00001d008909_T002 | 8 | 24852747 | 24855080 | - | 1011 | 336 | 6.55 | 35338.2 |
| ZmGA2ox12 | Zm00001d012712 | Zm00001d012712_T001 | 8 | 179086672 | 179088998 | - | 1017 | 338 | 5.48 | 35965.7 |
| ZmGA2ox12 | Zm00001d012712 | Zm00001d012712_T002 | 8 | 179086908 | 179088998 | - | 1017 | 338 | 5.48 | 35965.7 |
| ZmGA2ox13 | Zm00001d024175 | Zm00001d024175_T001 | 10 | 53027195 | 53031460 | - | 1116 | 371 | 7.79 | 39987 |
| ZmGA20ox1 | Zm00001d031926 | Zm00001d031926_T001 | 1 | 206982602 | 206985131 | - | 1323 | 440 | 5.91 | 47147.6 |
| ZmGA20ox2 | Zm00001d032223 | Zm00001d032223_T001 | 1 | 217837369 | 217838681 | - | 900 | 299 | 5.49 | 32358.7 |
| ZmGA20ox3 | Zm00001d034898 | Zm00001d034898_T001 | 1 | 305074531 | 305075830 | - | 1215 | 404 | 6.67 | 45000.9 |
| ZmGA20ox4 | Zm00001d003311 | Zm00001d003311_T001 | 2 | 39752127 | 39753847 | + | 903 | 300 | 5.32 | 32327.7 |
| ZmGA20ox5 | Zm00001d007894 | Zm00001d007894_T001 | 2 | 241897454 | 241898638 | - | 1185 | 394 | 7.26 | 43119.8 |
| ZmGA20ox6 | Zm00001d042611 | Zm00001d042611_T001 | 3 | 173559174 | 173562022 | - | 1161 | 386 | 6.52 | 42510.3 |
| ZmGA20ox7 | Zm00001d049926 | Zm00001d049926_T001 | 4 | 53429242 | 53431331 | - | 1212 | 403 | 6 | 43718.8 |
| ZmGA20ox8 | Zm00001d052999 | Zm00001d052999_T001 | 4 | 208285887 | 208290380 | + | 1332 | 443 | 7.94 | 49491.7 |
| ZmGA20ox9 | Zm00001d013725 | Zm00001d013725_T001 | 5 | 18631981 | 18633840 | + | 1050 | 349 | 5.5 | 39166.3 |
| ZmGA20ox9 | Zm00001d013725 | Zm00001d013725_T002 | 5 | 18632040 | 18633504 | + | 1116 | 371 | 5.51 | 40583.7 |
| ZmGA20ox9 | Zm00001d013725 | Zm00001d013725_T003 | 5 | 18632512 | 18633478 | + | 585 | 194 | 6.36 | 21546.3 |
| ZmGA20ox9 | Zm00001d013725 | Zm00001d013725_T004 | 5 | 18632631 | 18633478 | + | 555 | 184 | 5.97 | 21076.6 |
| ZmGA20ox10 | Zm00001d012212 | Zm00001d012212_T001 | 8 | 170115789 | 170118570 | - | 1392 | 463 | 8.53 | 50672.7 |
| ZmGA20ox11 | Zm00001d026431 | Zm00001d026431_T001 | 10 | 145720480 | 145722209 | - | 963 | 320 | 5.23 | 35373.9 |
| ZmGA3ox1 | Zm00001d039634 | Zm00001d039634_T001 | 3 | 9745656 | 9748061 | + | 1149 | 382 | 6.56 | 41510.5 |
| ZmGA3ox2 | Zm00001d037627 | Zm00001d037627_T001 | 6 | 132317697 | 132319277 | + | 1125 | 374 | 5.56 | 41160.1 |
| ZmGA3ox3 | Zm00001d018617 | Zm00001d018617_T001 | 7 | 1105512 | 1106576 | + | 1065 | 354 | 6.18 | 39155.2 |

genes were distributed in 10 chromosomes, except for chromosome 6. And chromosome 6 contained the most gibberellin-dioxygenases genes (6 genes). Further analysis showed that the gibberellin-dioxygenases genes only occurred fragment duplication while there were no tandem duplication events observed (**Fig 2**).

## Phylogenetic analysis of gibberellin-dioxygenases genes in maize

To investigate the phylogenetic relationships of the gibberellin-dioxygenases gene family in maize, 27 Gibberellin-dioxygenases genes in maize, together with 16 *Arabidopsis* and 22 rice

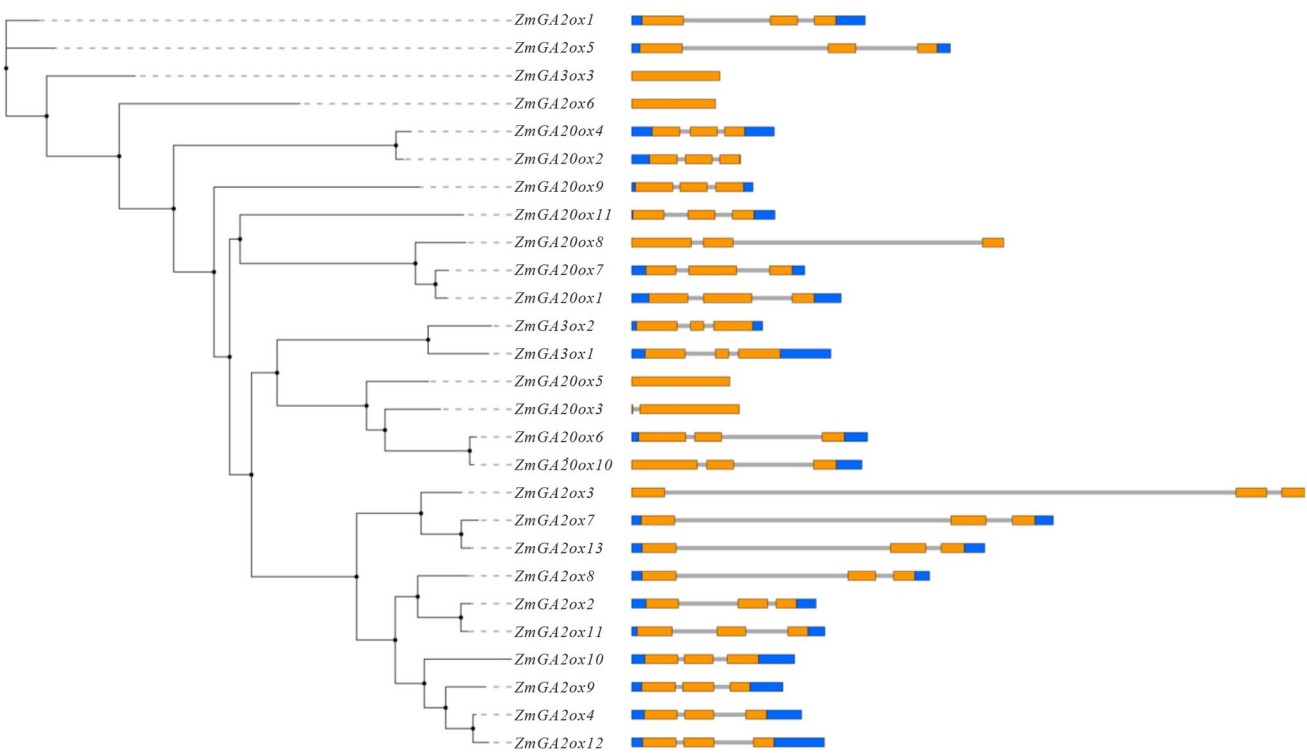

**Fig 1. Gene structure of gibberellin-dioxygenases genes in maize.** For gene structure analysis, genomic and CDS sequences were used for drawing gene structure schematic diagrams with the Gene Structure Display Server from the Center for Bioinformatics at Peking University (http://gsds.cbi.pku.edu.cn/index.php).

gibberellin-dioxygenases genes (**S2 Table**), were selected for phylogenetic analysis. As **Fig 3** shown, the gibberellin-dioxygenases proteins were clustered into seven groups, I to VII. There are 7, 3, 6, 4, 3, 2 and 2 gibberellin-dioxygenases genes were in I to IV groups, respectively. And further analysis showed that in every groups contained gibberellin-dioxygenases genes from Arabidopsis and rice, indicating that the differentiation of gibberellin-dioxygenases genes in maize is earlier than that of monocotyledonous and dicotyledonous plants.

## Conserved motifs analysis of gibberellin-dioxygenases genes in maize

The conserved motifs of gibberellin-dioxygenases protein sequences were further predicted using the MEME software. A total of 10 conserved motifs were found among all the gibberellin-dioxygenases genes (**Fig 4**). In consistent with phylogenetic tree of gibberellin-dioxygenases genes, the 27 gibberellin-dioxygenases genes were classified to 7 clades. Further analysis showed that all the gibberellin-dioxygenases proteins were lack of 1-motifs in I-IV clades. Of them, the proteins from clades I and II is lack of motif 9. The genes form Clade III, V and *ZmGA3ox2* from Clade VII showed the similar motifs constitution which lack of motif 9 and motif 8 except for *ZmGA3ox3* and *ZmGA2ox6* from Clade III and *ZmGA20ox8* from Clade V. The genes from clade IV and *ZmGA3ox1* from Clade VII showed the same motifs constitution which lack of motif 8 and motif 10. In addition, the *ZmGA20ox9* and from *ZmGA20ox9* showed the greatest degree of absence in conserved motifs that lack of 5 and 3 motifs, respectively. The results of the conserved motifs of gibberellin-dioxygenases genes showed evolutionary divergence in maize.

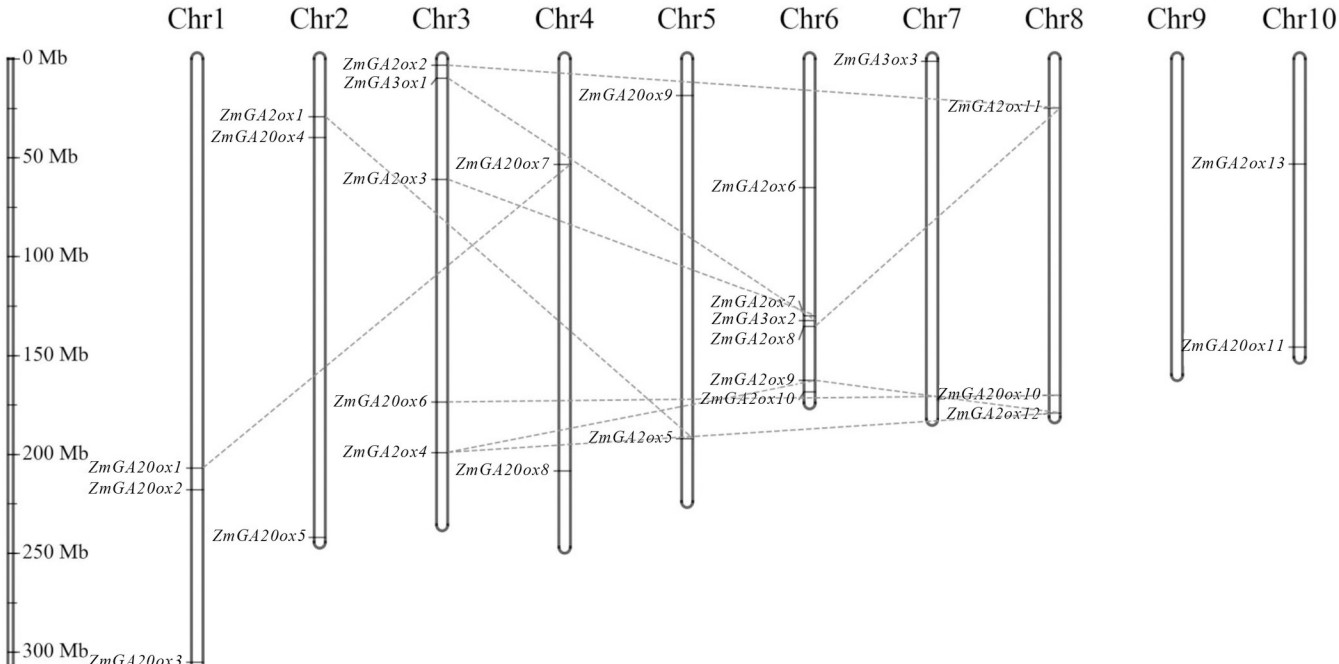

**Fig 2. Chromosome distribution for gibberellin-dioxygenases genes in maize.** The chromosome distribution was finished by MapGen2Chrom web V2(http://mg2c.
iask.in/mg2c_v2.0/).

## Tissue-specific expression profile analysis of gibberellin-dioxygenases genes in maize

We have demonstrated that the gibberellin-dioxygenases genes showed different conserved motifs. In order to insight into the putative functions of the gibberellin-dioxygenases genes in maize, the temporal and spatial expression profile of these identified gibberellin-dioxygenases genes were analyzed using the public RNA-Seq data (PRJNA314400) from different tissues (S3 Table). As **Fig 5** shown, the gibberellin-dioxygenases genes showed different tissue expression pattern in different tissues and most of the gibberellin-dioxygenases genes showed tissue specific expression. For example, *ZmGA3ox1* mainly expressed in the germinating period of kemels while *ZmGA3ox3* and *ZmGA20ox11* were mainly expressed in tip of the roots. In addition, we also found several gibberellin-dioxygenases were simultaneously expressed in the same tissue, such as *ZmGA20ox5*, *ZmGA20ox2* and *ZmGA2ox5* expressed in silks and *ZmGA20ox6*, *ZmGA20ox3*, *ZmGA20ox4* and *ZmGA20ox1*expressed in the transfer zone of matemal. The diversity of tissue expression pattern indicated the functional diversity of gibberellin-dioxygenases genes which will contribute to different morphogenesis in plant development.

## Expression analysis of gibberellin-dioxygenases genes responding to GA

A large number of gibberellin-dioxygenases genes have been demonstrated to regulate numbers of processes in response to GA. However, the studies focus on the gibberellin-dioxygenases in response to GA in maize is rare. Therefore, transcriptome of maize (PRJNA421076) in response to GA were used to explore the GA induced expression of gibberellin-dioxygenases genes in leaf and leaf sheath. As **Fig 6** shown, almost all the gibberellin-dioxygenases genes

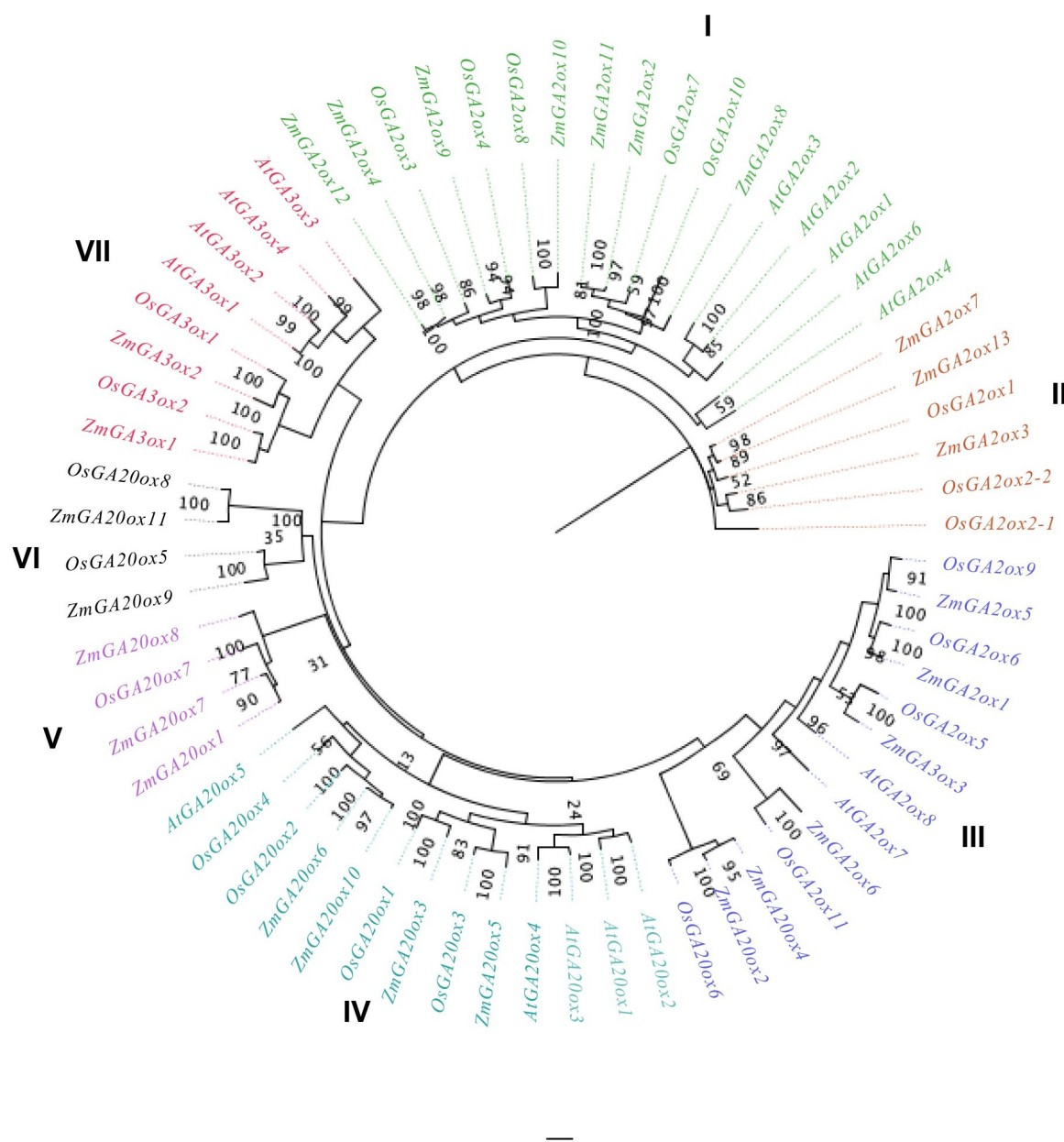

**Fig 3. Phylogenetic analysis of gibberellin-dioxygenases proteins among maize (27), Arabidopsis (16) and rice (22).** The phylogenetic tree was constructed based on the full-length protein sequences using Figtree software. Seven subgroups (I-VII) are shown in various colors.

were significantly elevated in response to GA except for *ZmGA2ox2* and *ZmGA20ox10* of 15 gibberellin-dioxygenases genes normally expressed in leaves. 10 and 11 gibberellin-dioxygenases genes showed up and down regulated under GA treatment than that under normal condition in leaf sheath. Further analysis showed that these differential expressed genes were from different groups, implying that gibberellin-dioxygenases genes might play different roles in response to GA.

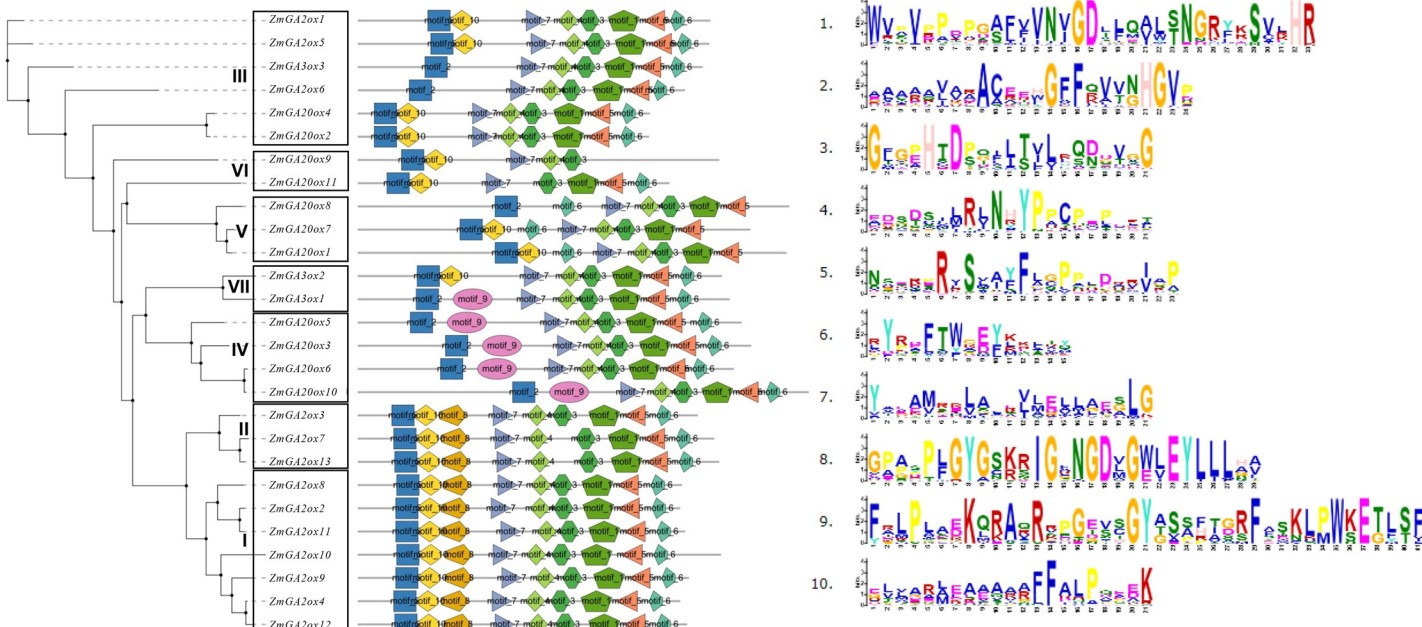

**Fig 4. Phylogenetic relationships and conserved motifs compositions of the 27 gibberellin-dioxygenases genes in maize.** MEME was used to predict conserved motifs. Each motif is represented by a different colored box. Amino acid sequences of gibberellin-dioxygenases genes were used to build the phylogenetic tree. Prottest was firstly use to predict the best evolution model and JTT+G+I+F as the best evolution model to build the evolution tree using RAxML 1000 bootstrap and the phylogenetic tree visualization is done using Figtree software.

## Candidate gibberellin-dioxygenases genes response to GA verified by qRT-PCR in maize

In order to explore the key GA stress-responsive candidates in maize, 6 Gibberellin-dioxygenasess based on the RNA-Seq data which showed the most significant upregulated in leaves or leaf sheath were selected to verified by qRT-PCR analysis at 6h, 12h, 24h, 48h and 72h after GA treatment. In consistent with the RNA-seq data, compared with control, the expression of *ZmGA2ox1*, *ZmGA2ox4*, *ZmGA20ox2* was significantly elevated in 6h and 24h, 6h and 12h, and 24h, respectively (Fig 7). However, the expression of Z*mGA20ox7*, *ZmGA3ox1* and *ZmGA3ox3* were significantly downregulated at all the times after GA treatment compared with control. This result may be caused by different varieties used in present. These results demonstrated *ZmGA2ox1*, *ZmGA2ox4* and *ZmGA20ox2* could consider to be key genes which played vital roles in GA stress.

## Discussion

Gibberellin (GA) is an essential hormone that is involved in many aspects of plant growth and development, including seed maturation, stem elongation and response to abiotic stress [22, 23]. Gibberellin-dioxygenases genes are reported to be involved in many critical development processes [24]. Systematic and integrative analyses of gibberellin-dioxygenases genes have been performed in Arabidopsis, rice and some other plants [5]. However, the gibberellin-dioxygenases genes in response to GA are less studied in maize compared with that in Arabidopsis and rice. Therefore, we sought to study the characteristics of this gene family in response to gibberellin by combining bioinformatic and expression analyses.

The details of how GAs is biosynthesis and deactivation have accumulated in the last few years and are beginning to explain in molecular terms the pleiotropic action of GA in plant

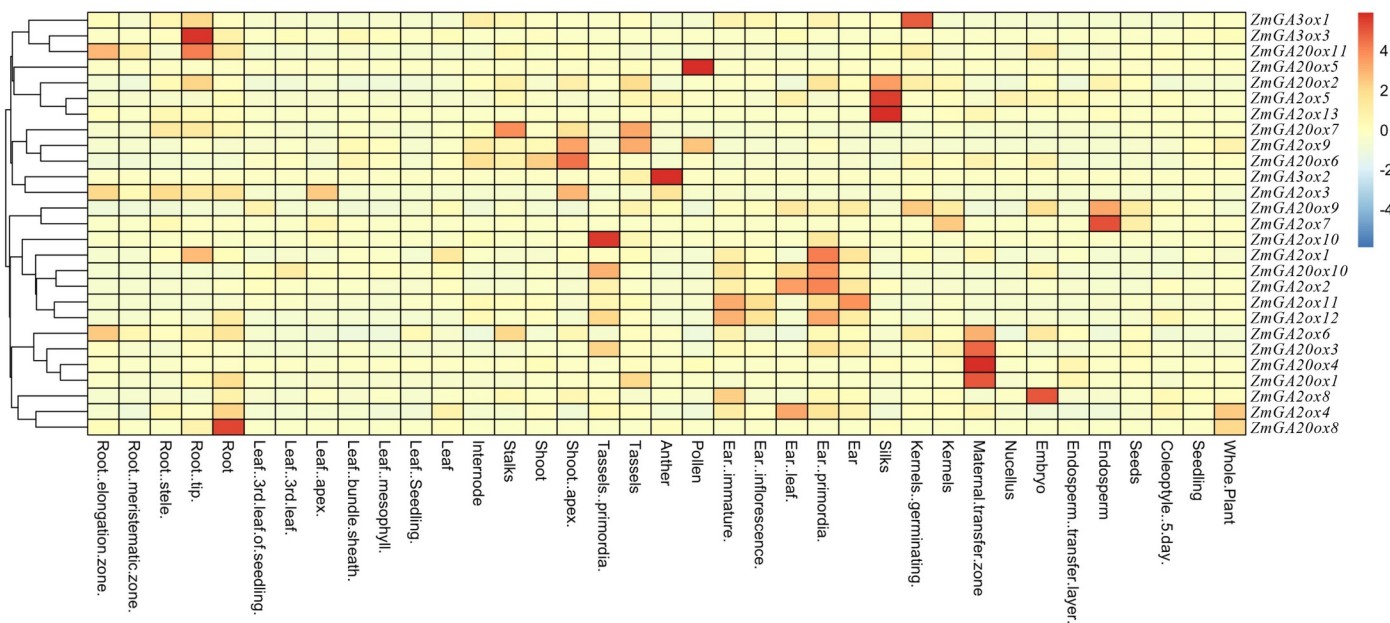

**Fig 5. Tissue-specific expression analysis of gibberellin-dioxygenases genes in maize.**

development [4, 5]. The 2-oxoglutarate dependent dioxygenases (2-ODDs), including GA20ox and GA3ox, are the key enzymes in a series of oxidation steps and, GA 2-oxidases GA2ox are the unique enzymes in the pathways and regulation of GA degradation [2]. And several gibberellin-dioxygenases genes were also investigated in maize, such as ga2ox1 [25]. In present study, 27 Gibberellin-dioxygenases genes were finally retained and used for further analysis, including 13 GA2ox1 genes, 11 GA20ox genes and 3 GA3ox (*ZmGA3ox1-3*) genes which is different from the numbers of other plants, such as 16 members in Arabidopsis thaliana [26], 21 members in rice [27], 24 members in soybean [8]. Gene duplications are considered to be one of the primary driving forces in the evolution of genomes and genetic systems [28]. Segmental and tandem duplications have been suggested to represent two of the main causes of gene family expansion in plants [29]. Further analysis showed that 27 gibberellin-dioxygenases genes were distributed in 10 chromosomes, except for chromosome 6. And gibberellin-dioxygenases genes only occurred segmental duplication while there were no tandem duplication events. These results are consistent with that segmental duplications multiple genes through polyploidy followed by chromosome rearrangements and occurs most frequently in plants because most plants are diploidized polyploids and retain numerous duplicated chromosomal blocks within their genomes [30]. Previous investigations of the gibberellin-dioxygenases genes in various plant species have divided the plant gibberellin-dioxygenases genes into different classes [5]. In present study, the gibberellin-dioxygenases proteins were clustered into seven groups, I to VII. And further analysis showed that in every groups contained gibberellin-dioxygenases genes from Arabidopsis and rice, indicating that the differentiation of gibberellin-dioxygenases genes in maize is earlier than that of monocotyledonous and dicotyledonous plants. Specific motifs in amino acid sequences are vital regions related to function. Previous analysis found that all the GA20ox, GA3ox and GA2ox sequences belonged to the 2-ODDs superfamily, which share high homology with the functional domains DIOX_N (PF14226) and 2OG-FeII_Oxy (PF03171). In consistent with phylogenetic tree of gibberellin-

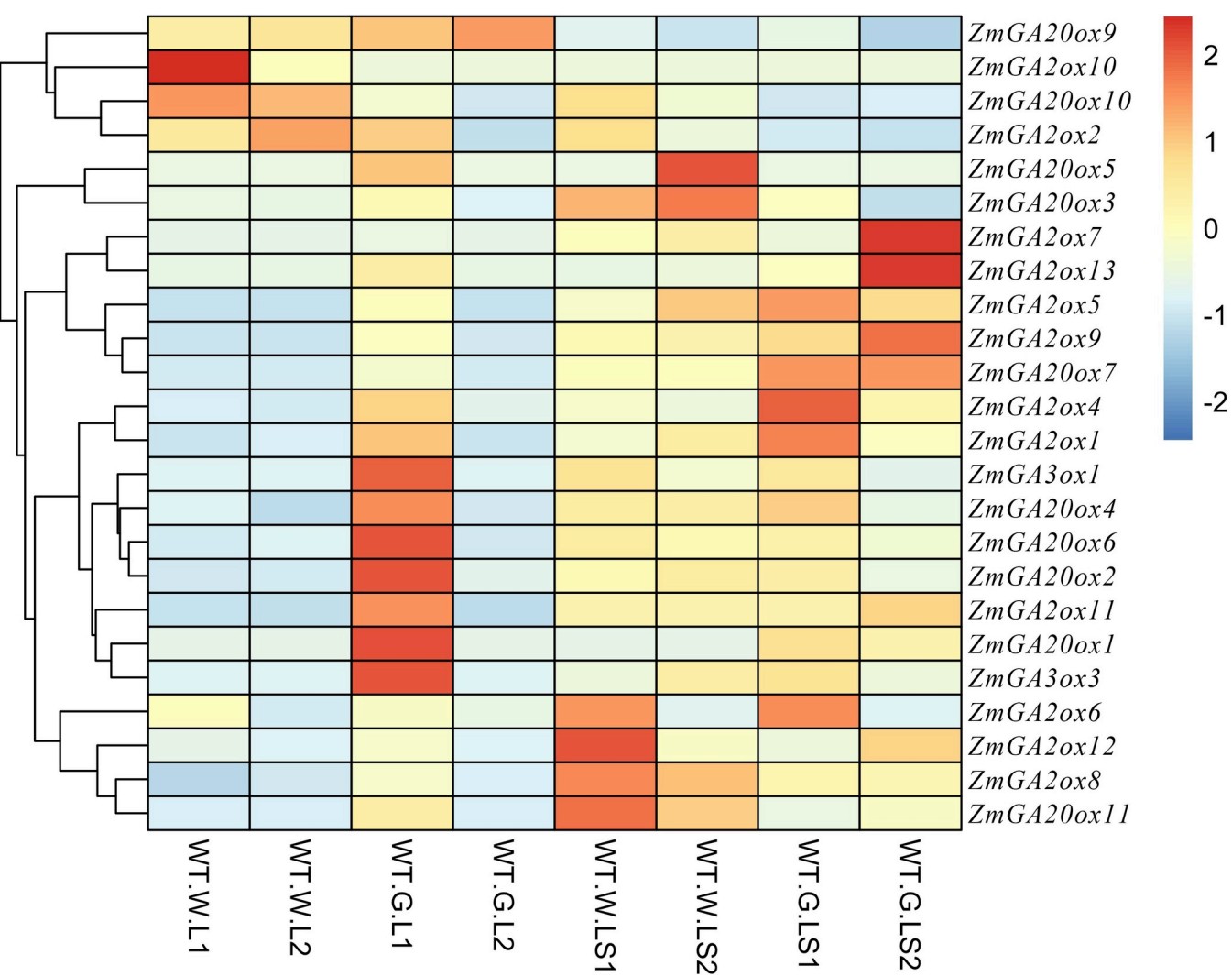

**Fig 6. Differential expressed genes in response to GA.**

dioxygenases genes, the 27 gibberellin-dioxygenases genes were classified to 7 clades. Further analysis showed that all the gibberellin-dioxygenases proteins were lack of 1–4 motifs in I-IV clades. The results of the conserved motifs of gibberellin-dioxygenases genes showed evolutionary divergence in maize, suggesting the divergent function in maize development of gibberellin-dioxygenases genes.

In order to investigated the divergent function caused by the conserved motifs, the expression of gibberellin-dioxygenases genes was investigated. The gibberellin-dioxygenases genes showed different tissue expression pattern in different tissues and most of the gibberellin-dioxygenases genes showed tissue specific expression. In fact, the gibberellin-dioxygenases genes from different plants have been studied [7]. And they played various kinds of functions in different plants, such as response to abiotic stress, increased biomass production and yield and plant development. For example, activation of gibberellin 2-oxidase 6 decreased active gibberellin levels and created a dominant semi-dwarf phenotype in rice (Oryza sativa L.) [31]. Overexpression of stga2ox1 gene increases the tolerance to abiotic stress in transgenic potato plants

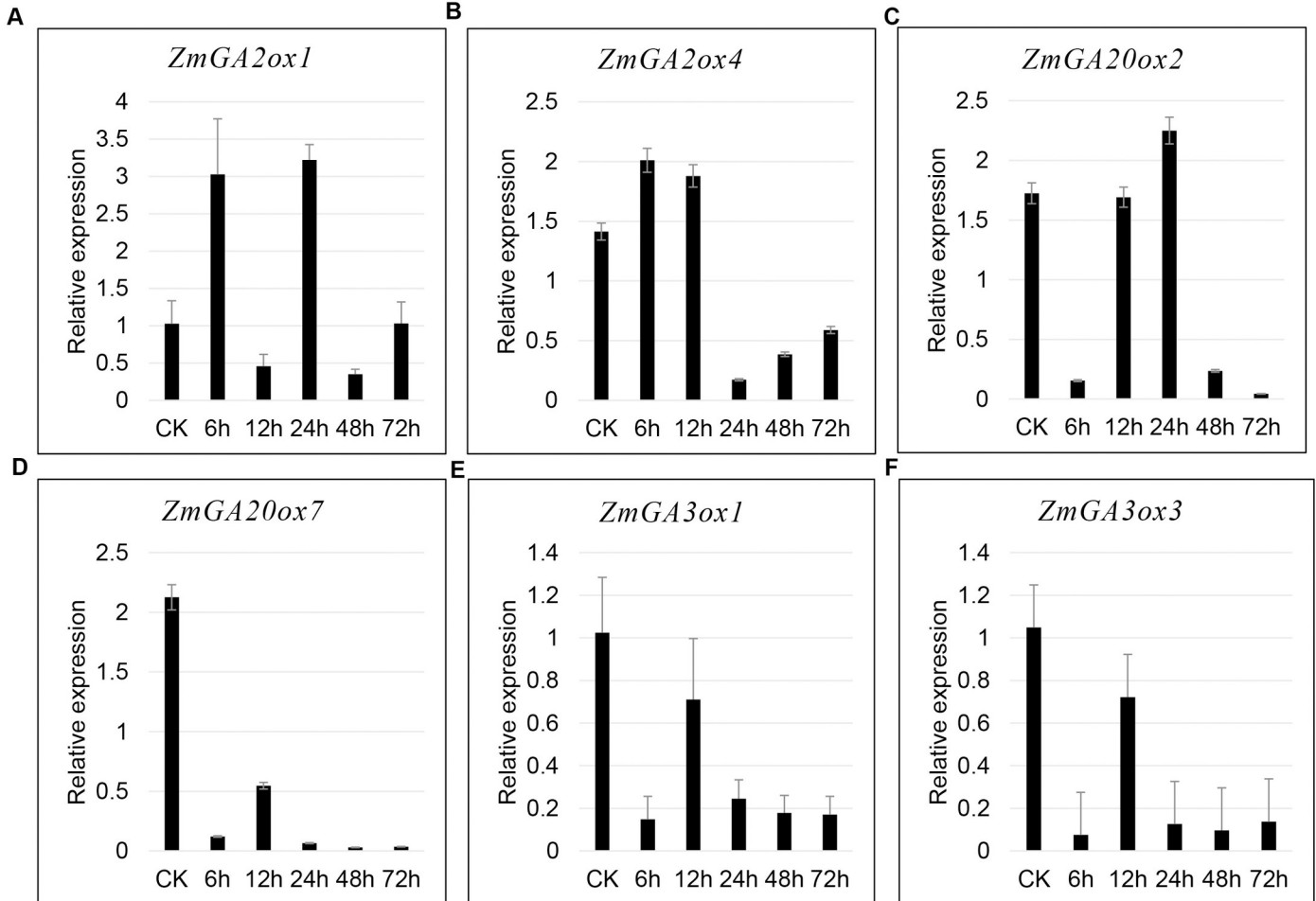

**Fig 7. The expression of candidate genes that were most significantly expressed in the response to GA using qRT-PCR analysis.** A-F, The expression of *ZmGA2ox1*, *ZmGA2ox4*, *ZmGA20ox7*, *ZmGA3ox1* and *ZmGA3ox3* after GA treatment, respectively. CK, Control check. **, $p < 0.01$, Student's t-test). Gene expression profiles were evaluated using the $2^{-\triangle\triangle Ct}$ methods.

[32]. Developing xylem-preferential expression of PdGA20ox1 improves woody biomass production in a hybrid poplar [33] and the QTL GNP1 encodes GA20ox1, which increases grain number and yield by increasing cytokinin activity in rice panicle meristems [34]. In addition, several gibberellin-dioxygenases genes have been clarified. Such as maize dominant dwarf11 (d11) mutant phenotype is related to the upregulation of GA20ox and GA2ox which contribute to gibberellin biosynthetic deficiency [18]. Muylle et al., demonstrated that overexpression of GA20-OXIDASE1 impacts plant height, biomass allocation and saccharification efficiency in maize [20]. And expression of *ZmGA20ox* cDNA alters plant morphology and increases biomass production of switchgrass (Panicum virgatum L.) [35]. Furthermore, it also showed that the maize transcription factor KNOTTED1 directly regulated the gibberellin catabolism gene ga2ox1 [25]. Taken together, the diversity of tissue expression pattern indicated the functional diversity of gibberellin-dioxygenases genes which will contribute to different morphogenesis in plant development and response to abiotic stress.

Wang et al., found some GA2ox, GA3ox, and GA20ox genes which showed differential expressed after GA treatment [19, 36]. In present study, almost all the gibberellin-dioxygenases genes were significantly elevated in response to GA except for *ZmGA2ox2 and ZmGA20ox10*

of 15 gibberellin-dioxygenases genes normally expressed in leaves. And 10 and 11 gibberellin-dioxygenases genes showed up and down regulated under GA treatment than that under normal condition in leaf sheath. Further analysis showed that these differential expressed genes were from different groups, implying that gibberellin-dioxygenases genes might play different roles in response to GA. Generally, in most plants, GA20ox and GA3ox which contribute to the production of bioactive GAs are downregulated by applied exogenous GA [37]. In contrast, the genes encoding GA2ox, which convert active GAs to inactive catabolites, are upregulated by GA treatment [38]. qRT-PCR results showed that compared with control, the expression of *ZmGA2ox1and ZmGA2ox4* was significantly elevated in 6h and 24h, 6h and 12h, respectively. However, the expression of *ZmGA20ox7*, *ZmGA3ox1* and *ZmGA3ox3* were significantly downregulated at all the times after GA treatment while *ZmGA20ox2* was significantly elevated at 24h compared with control. Our findings are in accordance with previous studies [39]. These results indicated that *ZmGA2ox1*, *ZmGA2ox4*, *ZmGA20ox7*, *ZmGA3ox1* and *ZmGA3ox3* might be potential genes for regulating balance of GAs which play essential roles in plant development. However, the precise function and mechanism of these candidate genes need to be further investigation.

## Conclusion

Our results provide a more comprehensive understanding of gibberellin-dioxygenases in maize, including phylogenetic analysis, gene structure and conserved motif characteristics, gene duplication and tissue expression. Totally, 27 Gibberellin-dioxygenases genes were identified which classified into seven subfamilies (I-VII) based on phylogenetic analysis, gene structure and conserved motif characteristics. And gibberellin-dioxygenases genes only occurred segmental duplication that occurs most frequently in plants. Furthermore, the diversity of tissue expression pattern indicated the functional diversity of gibberellin-dioxygenases genes which will contribute to different morphogenesis in plant development. Moreover, *ZmGA2ox1*, *ZmGA2ox4*, *ZmGA20ox7*, *ZmGA3ox1* and *ZmGA3ox3* were considered to be potential genes for regulating balance of GAs which play essential roles in plant development though transcriptome data and qRT-PCR. Our findings provided a basis for conducting in-depth mechanistic studies on the in distinct biological characteristics and adaptability in response to GA for gibberellin-dioxygenases genes in maize.

## Supporting information

**S1 Table. Primers used in present study.**
(DOCX)

**S2 Table. Gibberellin-dioxygenases genes in Arabidopsis, rice and maize.**
(XLSX)

**S3 Table. Tissue expression profiles for gibberellin-dioxygenases genes in maize.**
(XLSX)

## Acknowledgments

We are grateful to Wei Yang, Xueyu Cui, Liangyu Jiang and Xuejiao Ren for their contributions.

## Author Contributions

**Conceptualization:** Jiabin Ci, Xueyu Cui, Liangyu Jiang, Weiguang Yang.

**Data curation:** Xingyang Wang, Liangyu Jiang.

**Formal analysis:** Qi Wang, Fuxing Zhao, Xuejiao Ren.

**Software:** Wei Yang.

**Writing – original draft:** Jiabin Ci.

**Writing – review & editing:** Jiabin Ci.

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
