## [Decision Letter · Decision Letter 0]

12 Oct 2020

PONE-D-20-27828

Genome-wide analysis of the biosynthesis and deactivation of gibberellin-dioxygenases gene family and key genes identified in response to GA in maize

PLOS ONE

Dear Dr. yang,

Thank you for submitting your manuscript to PLOS ONE. After careful consideration, we feel that it has merit but does not fully meet PLOS ONE’s publication criteria as it currently stands. Therefore, we invite you to submit a revised version of the manuscript that addresses the points raised during the review process.

We look forward to receiving your revised manuscript.

Kind regards,

Keqiang Wu, Ph.D

Academic Editor

PLOS ONE

Journal Requirements:

The work was carried out with the financial support of The National Key Research and

Development Program of China（2016YFD0101202）; Science and Technology Development Plan

of Jilin Province ( 20190301013NY).

i) We note that you have provided funding information that is not currently declared in your Funding Statement. However, funding information should not appear in the Acknowledgments section or other areas of your manuscript. We will only publish funding information present in the Funding Statement section of the online submission form.

ii) Please remove any funding-related text from the manuscript and let us know how you would like to update your Funding Statement. Currently, your Funding Statement reads as follows:

 "No.The funders had no role in study design, data collection and analysis, decision to publish, or preparation of the manuscript."

 iii) Please include your amended statements within your cover letter; we will change the online submission form on your behalf.

4. Please include your tables as part of your main manuscript and remove the individual files. Please note that supplementary tables (should remain/ be uploaded) as separate "supporting information" files

Reviewers' comments:

Reviewer's Responses to Questions

**Comments to the Author**

1. Is the manuscript technically sound, and do the data support the conclusions?

Reviewer #1: Yes

Reviewer #2: Yes

2. Has the statistical analysis been performed appropriately and rigorously? 

Reviewer #1: Yes

Reviewer #2: N/A

3. Have the authors made all data underlying the findings in their manuscript fully available?

Reviewer #1: Yes

Reviewer #2: Yes

4. Is the manuscript presented in an intelligible fashion and written in standard English?

Reviewer #1: Yes

Reviewer #2: No

5. Review Comments to the Author

Reviewer #1: Ci et al. identified 27 gibberellin-dioxygenase proteins that are involved in GA biosynthesis and degradation in maize. The authors performed bioinformatic analysis and determined transcription levels of tested genes. Finally, five genes were speculated as potential genes for regulating GA concentrations, which may play essential roles in maize development. I only have minor comment to this study.

1. Abstract: line 14, “Gas” should be “GAs”. Please check the whole manuscript carefully and correct the mistake like this.

2. The names of all genes in the whole manuscript and figures should be italicized.

3. Part 2.1: the method is not described in sufficient detail. How did the authors collect control samples? Did you collect control samples after water treatment (no GAs) for 6, 12, 24, 48, and72 h? If so, why is this data not shown in Figure 7? If you only collected control sample at 0 h (before GA treatment), when you collected GA-treated samples at 6, 12, 24, 48, and72 h, you found some gene expressions were changed, how to exclude the photoperiod and temperature effect on these gene expression?

4. In addition, in figure 7, “CK” is the abbreviation of cytokinin in english, please correct.

5. Part 2.6: line 7: H2O should be H2O; line 10, what is the meaning of “XX”?

6. The third paragraph of the discussion is not enough because many gibberellin-dioxygenase genes have been functional studied in many other plants, such as Arabidopsis and rice.

7. Reference: some journal are full name, but some are abbreviation.

Reviewer #2: In this manuscript, the authors reported Genome-wide identification of gibberellin-dioxygenases gene family in maize. By employing sequence-based bioinformatic analysis and qRT-PCR based gene expression determination, 27 Gibberellin-dioxygenases genes were identified in maize. The similar research had been done a lot in other plant species. From this prospective, this study in maize is lack of novelty. Still, it represents a standard report in relevant fields. I would like to suggest an acceptance after revision.

1.In the context of reported data in this study, the title of this manuscript is of over-presentation. Should the following be better: Genome-wide analysis of gibberellin-dioxygenases gene family and their responses to GA applications in maize

2.Too much language flaws in tense, Singular/plural, and grammar. For instance in the abstract, Gibberellin-dioxygenases genes “plays”; In “the” present study; their “structures”; different expression “patterns”; tissue-specific, and so on. In the main text, for example, “In order to investigated”, wrong tense. Any way, it should be very easy to go through these by carefully self-reading. Even that, you may choose professional language editing services. IT MATTERS.

3.Some presentation are inexact, for instance:

(a)“ However, although much effort, the key genes and signaling pathways involved in GA remains elusive. “ The authors should define the context or put suitable limiting word in sentences like this one. In this case, at least the work in Arabidopsis, rice, and some others have accomplished a lot. Therefore, we know the key genes, the main pathways, and accordingly, we can refer to study their conserved counterparts in maize. So on so forth.

(b) “However, there is no systematic and complete investigation on gibberellin-dioxygenases genes family.” No, there have been quite a lot publications on different species. Should you please check and include as references? List below as examples:

Li, C., et al. (2019). "Comprehensive expression analysis of Arabidopsis GA2-oxidase genes and their functional insights." Plant Sci 285: 1-13.

Yan, J., et al. (2017). "Ectopic expression of GA 2-oxidase 6 from rapeseed (Brassica napus L.) causes dwarfism, late flowering and enhanced chlorophyll accumulation in Arabidopsis thaliana." Plant Physiol Biochem 111: 10-19.

Hu, Y.-X., Y.-B. Tao and Z.-F. Xu (2017). "Overexpression of Jatropha Gibberellin 2-oxidase 6 (JcGA2ox6) induces dwarfism and smaller leaves, flowers and fruits in Arabidopsis and Jatropha." Front Plant Sci 8: 2103.

Shan, X., Y. Li, et al. (2013). "Transcriptome Profile Analysis of Maize Seedlings in Response to High-salinity, Drought and Cold Stresses by Deep Sequencing." Plant Molecular Biology Reporter 31(6): 1485-1491.

Gou, J., S. H. Strauss, et al. (2010). "Gibberellins regulate lateral root formation in Populus through interactions with auxin and other hormones." Plant Cell 22(3): 623-639.

Yamaguchi, S. (2008). Gibberellin metabolism and its regulation. Annual Review of Plant Biology. Palo Alto, Annual Reviews. 59: 225-251.

(b)“the gibberellin-dioxygenases genes in response to GA are still poorly understood”, why? Too vague description.

6. PLOS authors have the option to publish the peer review history of their article (what does this mean?). If published, this will include your full peer review and any attached files.

Reviewer #1: No

Reviewer #2: No

---

## [Author Response · Author response to Decision Letter 0]

21 Feb 2021

Dear Editor:

On behalf of my co-authors, we thank you very much for giving us an opportunity to revise our manuscript, we appreciate editor and reviewers very much for their positive and constructive comments and suggestions. We have studied reviewer’s comments carefully and have made revision which marked in red in the paper. We have tried our best to revise our manuscript according to the comments. And the revised contents as follows:

Thanks for your comments, we have confirmed that our manuscript meets PLOS ONE's style requirements.

The work was carried out with the financial support of The National Key Research and

Development Program of China（2016YFD0101202）; Science and Technology Development Plan

of Jilin Province ( 20190301013NY).

i) We note that you have provided funding information that is not currently declared in your Funding Statement. However, funding information should not appear in the Acknowledgments section or other areas of your manuscript. We will only publish funding information present in the Funding Statement section of the online submission form.

Thanks for your comments, we have provided funding information in Funding Statement as follows: The work was carried out with the financial support of The National Key Research and Development Program of China（2016YFD0101202）; Science and Technology Development Plan of Jilin Province (20190301013NY).

ii) Please remove any funding-related text from the manuscript and let us know how you would like to update your Funding Statement. Currently, your Funding Statement reads as follows:

 "No. The funders had no role in study design, data collection and analysis, decision to publish, or preparation of the manuscript."

 iii) Please include your amended statements within your cover letter; we will change the online submission form on your behalf.

Thanks for your comments, the amended statements were added to the cover letter.

Thanks for your comments, we have established an ORCID iD and that it is validated in Editorial Manager.

4. Please include your tables as part of your main manuscript and remove the individual files. Please note that supplementary tables (should remain/ be uploaded) as separate "supporting information" files

Thanks for your comments, the tables have been included in our main manuscript.

Reviewers' comments:

Reviewer's Responses to Questions

Comments to the Author

1. Is the manuscript technically sound, and do the data support the conclusions?

Reviewer #1: Yes

Reviewer #2: Yes

2. Has the statistical analysis been performed appropriately and rigorously?

Reviewer #1: Yes

Reviewer #2: N/A

3. Have the authors made all data underlying the findings in their manuscript fully available?

Reviewer #1: Yes

Reviewer #2: Yes

4. Is the manuscript presented in an intelligible fashion and written in standard English?

Reviewer #1: Yes

Reviewer #2: No

5. Review Comments to the Author

Reviewer #1: Ci et al. identified 27 gibberellin-dioxygenase proteins that are involved in GA biosynthesis and degradation in maize. The authors performed bioinformatic analysis and determined transcription levels of tested genes. Finally, five genes were speculated as potential genes for regulating GA concentrations, which may play essential roles in maize development. I only have minor comment to this study.

1. Abstract: line 14, “Gas” should be “GAs”. Please check the whole manuscript carefully and correct the mistake like this.

Thanks for your comments, we have carefully checked the whole manuscript carefully and correct the mistakes in the manuscript.

2. The names of all genes in the whole manuscript and figures should be italicized.

Thanks for your comments, the names of all genes in the whole manuscript and figures have been italicized.

3. Part 2.1: the method is not described in sufficient detail. How did the authors collect control samples? Did you collect control samples after water treatment (no GAs) for 6, 12, 24, 48, and72 h? If so, why is this data not shown in Figure 7? If you only collected control sample at 0 h (before GA treatment), when you collected GA-treated samples at 6, 12, 24, 48, and72 h, you found some gene expressions were changed, how to exclude the photoperiod and temperature effect on these gene expression?

4. In addition, in figure 7, “CK” is the abbreviation of cytokinin in english, please correct.

Thanks for your comments, we have changed it in the manuscript.

5. Part 2.6: line 7: H2O should be H2O; line 10, what is the meaning of “XX”?

Thanks for your comments, we have revised it in the manuscript.

6. The third paragraph of the discussion is not enough because many gibberellin-dioxygenase genes have been functional studied in many other plants, such as Arabidopsis and rice.

Thanks for your comments, we have changed it in the manuscript.

7. Reference: some journal are full name, but some are abbreviation.

Thanks for your comments, we have revised the references as the format of PLOS one.

Reviewer #2: In this manuscript, the authors reported Genome-wide identification of gibberellin-dioxygenases gene family in maize. By employing sequence-based bioinformatic analysis and qRT-PCR based gene expression determination, 27 Gibberellin-dioxygenases genes were identified in maize. The similar research had been done a lot in other plant species. From this prospective, this study in maize is lack of novelty. Still, it represents a standard report in relevant fields. I would like to suggest an acceptance after revision.

1.In the context of reported data in this study, the title of this manuscript is of over-presentation. Should the following be better: Genome-wide analysis of gibberellin-dioxygenases gene family and their responses to GA applications in maize

Thanks for your comments, the title have been changed to “Genome-wide analysis of gibberellin-dioxygenases gene family and their responses to GA applications in maize”.

2.Too much language flaws in tense, Singular/plural, and grammar. For instance in the abstract, Gibberellin-dioxygenases genes “plays”; In “the” present study; their “structures”; different expression “patterns”; tissue-specific, and so on. In the main text, for example, “In order to investigated”, wrong tense. Any way, it should be very easy to go through these by carefully self-reading. Even that, you may choose professional language editing services. IT MATTERS.

Thanks for your comments, we have chosen professional language editing service to revised the manuscript.

3. Some presentation are inexact, for instance:

(a)“ However, although much effort, the key genes and signaling pathways involved in GA remains elusive. “ The authors should define the context or put suitable limiting word in sentences like this one. In this case, at least the work in Arabidopsis, rice, and some others have accomplished a lot. Therefore, we know the key genes, the main pathways, and accordingly, we can refer to study their conserved counterparts in maize. So on so forth.

Thanks for your comments, we have changed it in the manuscript.

(b) “However, there is no systematic and complete investigation on gibberellin-dioxygenases genes family.” No, there have been quite a lot publications on different species. Should you please check and include as references? List below as examples:

Li, C., et al. (2019). "Comprehensive expression analysis of Arabidopsis GA2-oxidase genes and their functional insights." Plant Sci 285: 1-13.

Yan, J., et al. (2017). "Ectopic expression of GA 2-oxidase 6 from rapeseed (Brassica napus L.) causes dwarfism, late flowering and enhanced chlorophyll accumulation in Arabidopsis thaliana." Plant Physiol Biochem 111: 10-19.

Hu, Y.-X., Y.-B. Tao and Z.-F. Xu (2017). "Overexpression of Jatropha Gibberellin 2-oxidase 6 (JcGA2ox6) induces dwarfism and smaller leaves, flowers and fruits in Arabidopsis and Jatropha." Front Plant Sci 8: 2103.

Shan, X., Y. Li, et al. (2013). "Transcriptome Profile Analysis of Maize Seedlings in Response to High-salinity, Drought and Cold Stresses by Deep Sequencing." Plant Molecular Biology Reporter 31(6): 1485-1491.

Gou, J., S. H. Strauss, et al. (2010). "Gibberellins regulate lateral root formation in Populus through interactions with auxin and other hormones." Plant Cell 22(3): 623-639.

Yamaguchi, S. (2008). Gibberellin metabolism and its regulation. Annual Review of Plant Biology. Palo Alto, Annual Reviews. 59: 225-251.

Thanks for your comments, we have added these references in the manuscript.

(b)“the gibberellin-dioxygenases genes in response to GA are still poorly understood”, why? Too vague description.

Thanks for your comments, we have changed it in the manuscript.

6. PLOS authors have the option to publish the peer review history of their article (what does this mean?). If published, this will include your full peer review and any attached files.

Do you want your identity to be public for this peer review? For information about this choice, including consent withdrawal, please see our Privacy Policy.

Reviewer #1: No

Reviewer #2: No

---

## [Decision Letter · Decision Letter 1]

5 Mar 2021

PONE-D-20-27828R1

Genome-wide analysis of gibberellin-dioxygenases gene family and their responses to GA applications in maize

PLOS ONE

Dear Dr. yang,

Thank you for submitting your manuscript to PLOS ONE. After careful consideration, we feel that it has merit but does not fully meet PLOS ONE’s publication criteria as it currently stands. Therefore, we invite you to submit a revised version of the manuscript that addresses the points raised by reviewer #1.

We look forward to receiving your revised manuscript.

Kind regards,

Keqiang Wu, Ph.D

Academic Editor

PLOS ONE

Journal Requirements:

Reviewers' comments:

Reviewer's Responses to Questions

**Comments to the Author**

1. If the authors have adequately addressed your comments raised in a previous round of review and you feel that this manuscript is now acceptable for publication, you may indicate that here to bypass the “Comments to the Author” section, enter your conflict of interest statement in the “Confidential to Editor” section, and submit your "Accept" recommendation.

Reviewer #1: (No Response)

Reviewer #2: All comments have been addressed

2. Is the manuscript technically sound, and do the data support the conclusions?

Reviewer #1: Yes

Reviewer #2: Yes

3. Has the statistical analysis been performed appropriately and rigorously? 

Reviewer #1: No

Reviewer #2: N/A

4. Have the authors made all data underlying the findings in their manuscript fully available?

Reviewer #1: Yes

Reviewer #2: Yes

5. Is the manuscript presented in an intelligible fashion and written in standard English?

Reviewer #1: Yes

Reviewer #2: Yes

6. Review Comments to the Author

Reviewer #1: I do not agree to accept the manuscript immediately unless the authors clearly answer my query. 1. In my previous comments, i asked how did the authors collect control samples in Part 2.1? The authors answered: “The seedlings without GA treatment at 6, 12, 24, 48, 72 h act as control.” If so, adding GA treated samples at 6, 12, 24, 48, 72 h, there should be ten samples. Therefore, ten expression data should be showed in Fig 7, but there is only one control and five GA treated samples in the revised manuscript. Why? How did the authors process the expression data? Please describe it in sufficient detail.

2.In my previous comments, i said: “in figure 7, “CK” is the abbreviation of cytokinin in english, please correct.” The authors answered “Thanks for your comments, we have changed it in the manuscript.” However, i can not see the change in figure 7 of the revised manuscript.

3.“There were at least three biological replicates for each experiment”? I want to know the exact number of replicates.

Reviewer #2: The authors answered some of their responses inadequately. But it does not matter. They did their best. This is a good work of reference to the maize community. At the end, I consider an acceptance to this manuscript.

7. PLOS authors have the option to publish the peer review history of their article (what does this mean?). If published, this will include your full peer review and any attached files.

Reviewer #1: No

Reviewer #2: No

---

## [Author Response · Author response to Decision Letter 1]

31 Mar 2021

On behalf of my co-authors, we thank you very much for giving us an opportunity to revise our manuscript, we appreciate editor and reviewers very much for their positive and constructive comments and suggestions. We have studied reviewer’s comments carefully and have made revision which marked in red in the paper. We have tried our best to revise our manuscript according to the comments. And the revised contents as follows:

Reviewer #1: I do not agree to accept the manuscript immediately unless the authors clearly answer my query. 1. In my previous comments, i asked how did the authors collect control samples in Part 2.1? The authors answered: “The seedlings without GA treatment at 6, 12, 24, 48, 72 h act as control.” If so, adding GA treated samples at 6, 12, 24, 48, 72 h, there should be ten samples. Therefore, ten expression data should be showed in Fig 7, but there is only one control and five GA treated samples in the revised manuscript. Why? How did the authors process the expression data? Please describe it in sufficient detail.

Thanks for your comments, we have verified our data again, the seedlings without GA treatment at 0h act as control which presented as one control and five GA treated samples. And we have revised it in the revised manuscript.

2.In my previous comments, i said: “in figure 7, “CK” is the abbreviation of cytokinin in english, please correct.” The authors answered “Thanks for your comments, we have changed it in the manuscript.” However, i can not see the change in figure 7 of the revised manuscript.

Thanks for your comments, CK is the abbreviation of “Control check”not cytokinin.We have explain it in the figure legends of Fig.7.

3.“There were at least three biological replicates for each experiment”? I want to know the exact number of replicates.

Thanks for your comments, the exact number of replicates is three.

Reviewer #2: The authors answered some of their responses inadequately. But it does not matter. They did their best. This is a good work of reference to the maize community. At the end, I consider an acceptance to this manuscript.

Thanks for your comments, 

7. PLOS authors have the option to publish the peer review history of their article (what does this mean?). If published, this will include your full peer review and any attached files.

Do you want your identity to be public for this peer review? For information about this choice, including consent withdrawal, please see our Privacy Policy.

Reviewer #1: No

Reviewer #2: No

---

## [Editor Report · Decision Letter 2]

6 Apr 2021

Genome-wide analysis of gibberellin-dioxygenases gene family and their responses to GA applications in maize

PONE-D-20-27828R2

Dear Dr. yang,

We’re pleased to inform you that your manuscript has been judged scientifically suitable for publication and will be formally accepted for publication once it meets all outstanding technical requirements.

Kind regards,

Keqiang Wu, Ph.D

Academic Editor

PLOS ONE
---

## [Editor Report · Acceptance letter]

13 Apr 2021

PONE-D-20-27828R2 

Genome-wide analysis of gibberellin-dioxygenases gene family and their responses to GA applications in maize 

Dear Dr. Yang:

I'm pleased to inform you that your manuscript has been deemed suitable for publication in PLOS ONE. Congratulations! Your manuscript is now with our production department. 

Kind regards, 

on behalf of

Professor Keqiang Wu 

Academic Editor

PLOS ONE